# Visceral Leishmaniasis Urbanization in the Brazilian Amazon Is Supported by Significantly Higher Infection Transmission Rates Than in Rural Area

**DOI:** 10.3390/microorganisms10112188

**Published:** 2022-11-04

**Authors:** Rodrigo R. Furtado, Ana Camila Alves, Luciana V. R. Lima, Thiago Vasconcelos dos Santos, Marliane B. Campos, Patrícia Karla S. Ramos, Claudia Maria C. Gomes, Márcia D. Laurenti, Vânia Lucia da Matta, Carlos Eduardo Corbett, Fernando T. Silveira

**Affiliations:** 1Parasitology Department, Evandro Chagas Institute (Secretary of Science, Technology and Innovation, Ministry of Health), Ananindeua 67030-000, Pará, Brazil; 2Pathology Department, Medical School of São Paulo University, São Paulo 1246-903, Paulo, Brazil; 3Tropical Medicine Nucleus, Federal University of Pará, Belém 66055-240, Pará, Brazil

**Keywords:** *Leishmania* (*L.*) *infantum chagasi*, human infection, prevalence and incidence, clinical-immunological profiles, urban and rural scenarios, Brazilian Amazon

## Abstract

This was an open cohort prospective study (2016–2018) that analyzed the prevalence and incidence rates of human *Leishmania* (*L.*) *infantum chagasi*-infection and the evolution of their clinical-immunological profiles in distinct urban and rural scenarios of American visceral leishmaniasis (AVL) in Pará State, in the Brazilian Amazon. These infection profiles were based on species-specific DTH/IFAT-IgG assays and clinical evaluation of infected individuals, comprising five profiles: three asymptomatic, Asymptomatic Infection [AI], Subclinical Resistant Infection [SRI], and Indeterminate Initial Infection [III]; and two symptomatic, Subclinical Oligosymptomatic Infection [SOI] and Symptomatic Infection [SI = AVL]. The two distinct scenarios (900 km away) were the urban area of Conceição do Araguaia municipality and the rural area of Bujaru municipality in the southeast and northeast of Pará State. Human populations were chosen based on a simple convenience sampling design (5–10% in each setting), with 1723 individuals (5.3%) of the population (32,464) in the urban area and 1568 individuals (8.9%) of the population (17,596) in the rural one. A serological survey (IFAT-IgG) of canine infection was also performed in both scenarios: 195 dogs in the urban area and 381 in the rural one. Prevalence and incidence rates of human infection were higher in the urban area (20.3% and 13.6/100 person-years [py]) than in the rural setting (14.1% and 6.8/100-py). The AI profile was the most prevalent and incident in both urban (13.4% and 8.1/100-py) and rural (8.3% and 4.2/100-py) scenarios, but with higher rates in the former. An III profile case evolved to SOI profile after four weeks of incubation and another to SI (=AVL) after six. The prevalence of canine infection in an urban setting (39.2%) was also higher (*p* < 0.05) than that (32%) in the rural zone. AVL urbanization in Pará State, in the Brazilian Amazon, has led to infection rates significantly higher than those in rural sites, requiring more intense control measures.

## 1. Introduction

According to the World Health Organization (WHO), visceral and cutaneous leishmaniasis (VL and CL) are among the main six global endemic diseases, with approximately 50,000–90,000 new VL cases and more than 1 million new CL cases annually [1]. VL and CL appear to be the most important emerging infectious diseases in the world, approaching 12 million cases with approximately 360 million people living in risk areas [2]. VL is an anthropozoonotic disease that affects populations on five continents. In Latin America, it is known as American visceral leishmaniasis (AVL) or “neotropical calazar” [3,4].

In terms of AVL etiology, our support for the New World parasite autochthony theory—*Leishmania* (*L.*) *infantum chagasi* Lainson and Shaw 2005 (=*Leishmania chagasi* Cunha and Chagas 1937)—should be clear not only due to prior evidence [3,5,6,7,8] but also due to new findings concerning molecular clock analysis with DNA polymerase alpha subunit gene (a highly conserved genomic region related to the evolutionary process of leishmanine parasites) showing that *L.* (*L.*) *infantum chagasi* from Honduras (Central America) proved to be considerably more ancestral (382.800 *ya*) than *L.* (*L.*) *infantum chagasi* (143.300 *ya*) from Brazil (South America) and *L.* (*L.*) *infantum* (13.000 *ya*) from Europe [9]. We adopted throughout this work the nomenclature of the subspecies, *Leishmania* (*L.*) *infantum chagasi*, whose transmission in the New World is mainly through infected females of the phlebotomine vector, *Lutzomyia longipalpis* [10].

With regard to the clinical-immunological manifestations of human *L.* (*L.*) *infantum chagasi*-infection, it is important to note that despite the fact that AVL represents the most important expression from the medical point of view, other infection stages have been identified in recent studies carried out in the Brazilian Amazon, which are also part of the clinical-immunological spectrum of infection. In this sense, the combined use of semi-quantitative delayed-type hypersensitivity (DTH) and an indirect fluorescent antibody test (IFAT-IgG) associated with the clinical status of infected individuals have allowed the diagnosis of the broadest clinical-immunological spectrum of infection composed of five profiles: three asymptomatic ones, (1) Asymptomatic Infection (AI = DTH^+/++++^, IFAT^−^), (2) Subclinical Resistant Infection (SRI = DTH^+/++++^, IFAT^+/++^), and (3) Indeterminate Initial Infection (III = DTH^−^, IFAT^+/++^); and two symptomatic ones, (4) Symptomatic Infection (SI = AVL) and, (5) Subclinical Oligosymptomatic Infection (SOI), both with the same immune profile (DTH^−^, IFAT^+++/++++^). Of these, three (AI, SI [=AVL], and SOI) were previously identified and also confirmed [11,12,13], in addition to two newly diagnosed by this methodological approach [14]—expanding the clinical-immunological spectrum of human *L.* (*L.*) *infantum chagasi*-infection in Brazil. Subsequently, other studies of a prospective nature carried out in the Brazilian Amazon focusing on the dynamics of the clinical-immunological evolution of the infection estimated that 1–3% of III profile cases evolve to AVL (=SI profile) [15,16]. More recently, the transcriptome analysis of human *L.* (*L.*) *infantum chagasi*-infection in the Brazilian Amazon has confirmed that each clinical-immunological profile of infection has a specific transcriptional gene expression that can determine the outcome of infection [17].

In Brazil, AVL is admittedly a zoonosis of a typically rural character; however, in the last two decades, the disease has changed its epidemiological profile in order to advance to medium and large urban areas and, consequently, transformed into a disease with a large urban epidemiological profile, most likely originated by anthropogenic environmental and demographic processes [18,19,20,21]. Therefore, it has become increasingly important to examine the dynamics of human *L.* (*L.*) *infantum chagasi*-infection not only in its original rural environment but also in recent urban situations in order to develop new strategies to control its expansion.

The main objective of this open cohort prospective study was, therefore, to evaluate the prevalence and incidence rates of human symptomatic and asymptomatic *L.* (*L.*) *infantum chagasi*-infections, examine the evolution of their clinical-immunological profiles (SI = AVL, SOI, III, SRI, and AI) in distinct urban and rural scenarios in the Brazilian Amazon, and focus greater attention on public policies of surveillance, control, and infection prevention that could optimize early AVL diagnosis and treatment.

## 2. Materials and Methods

### 2.1. Study Area

The present study was carried out in two distinct epidemiological scenarios in the Brazilian Amazon: the urban area of Conceição do Araguaia municipality in southeastern Pará State (49°15′53″ S: 49°35′53″ W) and the rural Bujaru municipality in northeastern Pará State (01°30′54″ S: 48°02′41″ W). The two areas are approximately 900 km apart (Figure 1). The climate in both is typically equatorial, with a mean temperature of 28 °C and high humidity. Up until the first decade of the present century, Conceição do Araguaia municipality was considered free of AVL, while Bujaru has been experiencing AVL for more than 30 years when infections strongly expanded into northern and northeastern Pará State. During the most recent six-year period [2012–2017], however, the average numbers of AVL cases in Conceição do Araguaia municipality (19.3) were more than twice those of Bujaru (7.8) [22].

### 2.2. Study Population

The present study included 1723 individuals (5.3%) of the total population of the urban area of Conceição do Araguaia municipality (total population 32,464) and 1568 individuals (8.9%) of four small villages in the rural area of Bujaru municipality (total population 17,596) [23]. The 1723 individuals from the urban area were distributed among the four of the city’s five neighborhoods: Centro (452), Emerêncio (434), Vila Nova (423), and Vila Cruzeiro (414). The 1568 individuals from the rural area were distributed among four small villages: Ponta de Terra (567), São Pedro (372), Mariahí (318), and Tracuateua-Trindade (311) (Figure 1).

The participating populations in both epidemiological scenarios were chosen based on a simple convenience sampling design, but with an effort to account for 5–10% of the target population in each setting, that is, 5.3% (1723 individuals) of the total population (32,464) of the urban area, and 8.9% (1568 individuals) of four small villages in the rural one (17,596 total population). We attempted to enlist a representative sample from each of the neighborhoods in the urban area and from each village in the rural setting and to match those samples by the numbers of families in each area (354 and 388, respectively). All families included in the study denied any previous occurrence of Chagas disease in their relatives, and all of the neighborhoods in the urban setting and all of the villages in the rural setting had reported AVL cases in the prior two years.

The enrolled populations were composed of both men and women, one-year-old or older. Those subjects were divided into three age groups: 1–10, 11–20, and ≥21 years. Of the 1723 individuals examined in the urban area, 702 (40.7%) were males, and 1021 (59.3%) were females, with ages ranging from 1 to 95 years (average age 30.3), with the following age distributions: 1–10 years (22.4%; *n* = 386), 11–20 years (19.8%; *n* = 341), and ≥21 years (57.8%; *n* = 996). Of the 1568 individuals examined in the rural area, 759 (48.4%) were males and 809 (51.6%) females, with ages varying from 1 to 93 (average age 26.7) and with the following age distribution: 1–10 years (24.3%; *n* = 381), 11–20 years (22.5%; *n* = 353), and ≥21 years (53.2%; *n* = 834). Any individuals diagnosed with AVL (=SI profile) and undergoing treatment were excluded from the present study.

### 2.3. Study Design

This was an open cohort prospective study designed to evaluate the prevalence and incidence rates of human *L.* (*L.*) *infantum chagasi*-infection and the evolution of their clinical-immunological profiles (SI = AVL, SOI, III, SRI, and AI) in distinct urban (2016–2017) and rural (2017–2018) scenarios in the Brazilian Amazon. The study identified the clinical-immunological infection profiles in the two settings (mainly profile III) with a subsequent follow-up period of one year in light of the high significance of III evolution towards either the resistance (AI profile) or susceptibility (SI profile = AVL) poles of infection. However, it is necessary to clarify that to assess the prevalence and incidence rates of infection, the numbers initially presented in the study population section above refer to the global population of the study (1723 in the urban area and 1568 in the rural area) and include the initial subject populations (1211 and 1329 respectively) plus the addition of new participants who entered the second stage of the study (512 in the urban area and 239 in the rural area). To determine the prevalence rates, we used only the initial population as the denominator in both areas. To calculate the incidence rates, however, we considered the person-years at risk, as this was an open cohort prospective study. In this case, one person-year at risk refers to a person who was accompanied for at least one year. The people who participated only in the second stage of the study each corresponded to 0.5 person-years at risk. To calculate the incidence rates, we used the total person-years at risk as the denominator, and the results were expressed as incidence rate per 100 person-years (py) (Figure 2).

The criteria for discriminating human cases of infection and the clinical-immunological profiles, as well as the susceptible or resistant cases of the III profile, were described in previous studies [14,15,16] but can be summarized as follows: cases of infection were characterized by the presence of reactivity with DTH, IFAT-IgG, or both. The clinical-immunological profiles are modulated by simultaneous analysis of DTH and IFAT-IgG reactivity in such a way that the AI profile is characterized by positive DTH reactivity and the absence of an IgG-antibody response (below the IFAT-IgG dilution cut-off of 1:80)—which seems to be strongly linked to genetic resistance to infection [24]. The SI profile (=AVL), on the other hand, is strongly linked to genetic susceptibility to infection, with clear DTH inhibition and the high expression of an IgG-antibody response. The SOI and SRI profiles represent borderline genetic expressions of disease susceptibility and resistance, respectively; the former generally evolves into mild clinical signs of susceptibility [such as fever, asthenia, cutaneous pallor, and moderate enlargement of the spleen] but with spontaneous clinical infection resolution after one to two months [25]; the latter profile presents an asymptomatic stage evolving towards the AI resistance profile. The asymptomatic III profile represents the earliest stage of infection [which is not well-defined from an immune-genetic point of view], with the capacity to evolve to either the AI-resistant profile or to the SI-susceptible profile (=AVL). Therefore, depending on the immune-genetic background of the infected individuals, infection can evolve to the AI-resistant profile or to the SI-susceptible profile (=AVL) after first passing through either the SRI or SOI profile, respectively [14,15,16].

### 2.4. Clinical Evaluation of Infected Individuals in Distinct Urban and Rural Scenarios

All individuals in the urban and rural settings presenting any type of immune response by DTH or IFAT-IgG (or both) were clinically examined (through a complete physical examination) to identify any signs and/or symptoms that could be recognized as classical features of SI (=AVL) [daily fever lasting up to two months, weakness, indisposition, cough, loss of appetite, weight loss, skin-mucous pallor, diarrhea, abdominal distension, hepatosplenomegaly] or of SOI [fever, asthenia, cutaneous pallor, and moderate enlargement of the spleen] profiles [4,13,25,26]. Only those cases presenting typical features of AVL [daily fever, weakness, weight loss, and hepatosplenomegaly] and hematological changes [anemia, leukopenia, and thrombocytopenia], however, received conventional antimony therapy, as recommended by the Brazilian AVL Control Program [27]; SOI profile cases with mild or moderate symptoms were monitored biweekly for two months period through clinical (with a complete physical examination) and laboratory (IFAT-IgG assay and blood count) evaluations [25].

Recognizing, however, that III profile cases are those that, depending on the patient’s immune-genetic background, can potentially evolve to the infection resistance pole (AI profile) or (principally) to the susceptibility pole (SI profile = AVL), all III cases were examined weekly for periods of up to three months, considering the probable infection incubation period [13]. They also underwent serological screening for immunoglobulin M (IFAT-IgM), as it has been demonstrated that III cases presenting IgM reactivity show high susceptibility to AVL—indicating that IgM reactivity in asymptomatic III cases seems to be a pivotal marker for AVL development [28,29,30].

All clinical evaluations were carried out by medical professionals at health centers in each neighborhood in the urban area; individuals in the rural area were scheduled for medical care at the municipal hospital located in the municipal center. Only infected individuals with an III profile were followed for a period of three months; those having AI or SRI profiles underwent only one clinical screening, as they already showed reactivity to infection resistance marker (DTH) [24].

### 2.5. Serological Diagnostic Survey of Canine L. (L.) Infantum Chagasi-Infection in Distinct Urban and Rural Scenarios

Considering that the presence of canine *L.* (*L.*) *infantum chagasi*-infection represents an unquestionable risk factor for the onset of human infections and, consequently, of AVL (=SI profile), a serological diagnostic survey of canine infections was performed in each epidemiological scenario (urban and rural). That survey was undertaken simultaneously with the survey of human infection prevalence to correlate the prevalence of canine infections with those of humans. To that end, the serum samples of 195 dogs in the urban setting and 381 dogs in the rural area were examined using the IFAT-IgG assay, according to previous work [31].

### 2.6. Spatial Distribution of Human L. (L.) Infantum Chagasi-Infection in Distinct Urban and Rural Scenarios

Positive cases of human *L.* (*L.*) *infantum chagasi*-infection were regarded as spatial units of infection, and their distributions were georeferenced in both the urban and rural settings using a handheld GPS (Garmin 76CSx). The patterns of case distributions were analyzed using dot maps with spatial resolutions at the community (rural setting) and neighborhood levels (urban setting) using QGIS version 3.10 software. Data source files were provided by the “Instituto Brasileiro de Geografia e Estatistica (IBGE)” and satellite images by Google Earth.

### 2.7. Data Analysis

The chi-square and Fisher’s exact tests were used to determine the significance of the differences between the prevalence and incidence rates, as well as the clinical-immunological infection profiles in the distinct urban and rural epidemiological scenarios, at a 5% level of significance, with 95% confidence intervals, using Bio-Estat 5.0 software [32].

### 2.8. Ethical Approval

The objective of the present study was presented to each individual surveyed, and only those who agreed to participate were included in the study by signing an informed consent form. In the case of the participation of minors, the consent of their parents or guardians was also requested by signing an informed consent form. The study was approved by the Research Programs Evaluating Committee of the Faculty of Medicine, University of São Paulo, São Paulo State, Brazil, under protocol number 127115/2016, and with approval number 3.156.918.

## 3. Results

### 3.1. Prevalence of Human L. (L.) Infantum Chagasi-Infection in Distinct Urban and Rural Scenarios

Among the 1211 individuals examined during the prevalence survey in the urban setting in 2016, 246 cases of infection (169 by DTH, 40 by IFAT-IgG, and 37 by both) were detected. Of the 1329 individuals examined in the rural setting in 2017, 188 cases of infection (116 by DTH, 42 by IFAT, and 30 by both) were detected. The prevalence rate of 20.3% (95% CI: 18.0–22.6%) in the urban setting was, therefore, higher (*p* < 0.05) than the 14.1% rate in the rural setting (95% CI: 12.3–16.0%).

### 3.2. Incidence of Human L. (L.) Infantum Chagasi-Infections in Distinct Urban and Rural Scenarios

In order to assess the incidence rates of human infection, we enrolled 512 individuals from the urban area and 239 from the rural one who took part as person-years (py) at risk in the second stage of the study. Their inclusion generated 1007 individuals for the denominator of the urban area and 1107 individuals for the denominator of the rural one, accounting for the losses of 214 (22.2%) and 153 (13.4%) individuals from both areas due to their refusal to continue participating in the study or because they could no longer the found in either area. A total of 137 new cases of infection were diagnosed (97 by DTH, 18 by IFAT-IgG, and 22 by both) among 1007 individuals in the urban area, while 75 new cases of infection were diagnosed (48 by DTH, 15 by IFAT-IgG, and 12 by both) among 1107 individuals in the rural one, revealing a higher incidence rate (*p* < 0.05) of 13.6/100-py (95% CI: 10.8–16.4%) in the urban area than the 6.8/100-py in the rural one (95% CI: 4.8–8.7%).

### 3.3. Frequency of Human L. (L.) Infantum Chagasi-Infections According to Age and Gender in Distinct Urban and Rural Scenarios

No differences were found in terms of the distributions of human infections by age (*p* > 0.05) in the two under-21-year-old groups (1–10 and 11–20) in the urban (12.7% (49/386) and 20.2% (69/341), respectively) and rural (8.6% (33/381) and 19.8% (70/353)) settings. A significant difference was observed (*p * < 0.05), however, between the adult rate (≥21 years) of 26.6% (265/996) in the urban setting as compared to the rural setting, 19.2% (160/834).

In terms of the distribution of human infections by gender, the infection rates of males in the urban (26.2%; 184/703) and in the rural settings (20.3%; 154/759) were both higher (*p* < 0.05) than those of females in the urban (19.5%; 199/1020) and in the rural settings (13.5%; 109/809).

### 3.4. Prevalence and Incidence of the Clinical-Immunological Profiles of Human L. (L.) Infantum Chagasi-Infections in Distinct Urban and Rural Scenarios

The prevalence of the different clinical-immunological profiles of human infection showed that the AI profile was the most prevalent in both settings (urban 13.4% (162/1211) and rural 8.3% (110/1329)) and higher in the urban setting than in the rural one (*p* < 0.05). Following in decreasing order, and with no significant differences (*p* > 0.05) in prevalence rates between the two scenarios (urban and rural), the corresponding profiles may be seen in Table 1.

In terms of the incidences of those profiles, it was noted that the AI profile had a higher incidence rate (*p* < 0.05) in both scenarios (urban 9.6/100 py (97/1007) and rural 4.3/100 pr (48/1107)), and that its incidence in the urban area (9.6/100 py) was greater (*p* < 0.05) than that (4.3/100 py) in the rural one. Additionally, following in that decreasing order and with no differences (*p* > 0.05) in incidence rates between the two scenarios, the corresponding clinical-immunological profiles are presented in Table 2.

In addition to the prevalence and incidence rates of the clinical-immunological infection profiles, those variables were also analyzed according to gender and age range (Table 3 and Table 4), with emphasis on those showing significant differences (*p* < 0.05). Regarding prevalence in terms of gender, a higher AI profile rate was observed in males (5.0%) than in females (3.3%) in rural area. Regarding prevalence in terms of age, higher rates of the AI, SRI, and III profiles were observed in the urban area in the age range ≥21 years (8.8%, 1.9%, and 2.0%, respectively) than in the age ranges of 11–20 (2.6%, 1.2%, and 0.5%) or 1–10 (2.1%, 0.3%, and 0.5%). Similar results were observed in terms of the AI and SRI profiles in the rural area, as they were higher in the age range≥ 21 years (5.2% and 1.7%) than in the age ranges of 11–20 (2.4% and 0.5%) or 1–10 (0.7% and 0.3%) (Table 3).

In evaluating incidence (100-py) in terms of gender and/or age, it should be noted that the only significant result related to the age range variable (but only for the AI profile) was the incidence rates observed in the ≥21 years range, both in urban (7.4) and rural (2.9) areas; they were greater than those of the 11–20- and 1–10-year-old groups, in both urban (1.5 and 0.7, respectively) and rural (1.3 and 0.2) settings (Table 4).

### 3.5. Evolution of III Profile Cases of Human L. (L.) Infantum Chagasi-Infections in Distinct Urban and Rural Scenarios

Among the 52 III profile cases in the urban setting, the situation of a family of five attracted our attention: a mother (35 years old) and her four children (a 13-year-old girl and three boys (eleven, six, and two years old)). Everyone in the family showed immunological markers for infection; the mother, the girl, and two of the boys showed IFAT-IgG reactivity (with titers ranging from 80 to 320) but not DTH reactivity, characterizing the III profile; the other boy showed IFAT-IgG (80) as well as to DTH reactivity (15 × 15 mm), characterizing the SRI profile. To complete the first evaluation, IFAT-IgM serology was performed, as usual, in all four III cases, the results of which were negative for all. However, four weeks after this evaluation, the two-year-old boy presented mild to moderate daily hyperthermia (37.5–38 °C) accompanied by asthenia and lack of appetite for 18 days. It was then that a new IFAT-IgG test showed an increased titer (to 640) while nonetheless maintaining the IFAT-IgM negative. After those 18 symptomatic days, the boy’s clinical condition regressed spontaneously without showing clinical signs of relapse after three months of follow-up. It was concluded that this was an III profile case that evolved to an SOI profile and then regressed to a spontaneous clinical cure (which was confirmed one year later (2017) by DTH conversion) to assume the clinical-immunological status of a resistant SRI profile. The other 48 III cases likewise did not show IFAT-IgM reactivity and did not present any clinical signs of infectious activity.

Of the 42 III profile cases diagnosed in the rural site, another family evidenced the infection evolution of an III profile. In this case, the family consisted of an adult couple and two male adolescents, who were all negative for IFAT-IgG and DTH in the first evaluation (2017)–indicating that there had been no previous contact with the parasite. In the following year (2018), the mother and one of the adolescents showed reactivity only to IFAT-IgG (with 160 and 320 titers, respectively), characterizing an infection diagnosis of the III profile, while the father and the other adolescent remained negative for both IFAT-IgG and DTH. Soon after that diagnostic screening, both III profile cases were tested for IFAT-IgM, and only the adolescent showed reactivity (with a 160 titer); six weeks after that evaluation, he came to show clinical signs and symptoms suggestive of AVL (elevated hyperthermia [38–39 °C] daily, chills, asthenia, weakness, headache, cough, lack of appetite). A new serology then revealed increased titers (1280) of both IFAT-IgG/IgM, confirming an acute AVL diagnosis. Antimony therapy with 15 mg/Sb^v^/kg for twenty-five days promoted the remission of all clinical signs and symptoms. The mother, on the other hand, remained asymptomatic during the entire period (three months).

### 3.6. Serological Diagnostic Survey of Canine L. (L.) Infantum Chagasi-Infections in Distinct Urban and Rural Scenarios

The prevalence of canine infection was considered here as a correlation parameter with that of human infection. As such, the same IFAT-IgG serological method used for the diagnosis of human infection was also used for the serological diagnosis of canine infection, which showed a 39.2% prevalence rate of canine infection (76/194) in the urban setting, greater (*p* < 0.05) than the 32% prevalence (122/381) in the rural one.

### 3.7. Spatial Distribution of Human L. (L.) Infantum Chagasi-Infections in Distinct Urban and Rural Scenarios

The spatial distributions of human infections in the urban and rural scenarios showed distinct profiles: clusters of infection cases where contiguous among the four districts of the city of Conceição do Araguaia in the urban scenario, and it was not possible to isolate clusters by neighborhood (Figure 3); the rural scenario, on the other hand, showed a much sparser distribution of infection cases in four isolated areas or villages (with the exception of two neighboring areas, Ponta de Terra and São Pedro, which formed a single cluster) (Figure 4).

## 4. Discussion

This represents the first study undertaken in the Brazilian Amazon that evaluated the prevalence and incidence rates of human *L.* (*L.*) *infantum chagasi*-infections, as well as the evolution of their symptomatic (SI [=AVL] and SOI) and asymptomatic (AI, SRI, and III) clinical-immunological profiles in two distinct epidemiological scenarios, urban and rural, combining clinical, humoral (IFAT-IgG) and cellular (DTH skin reaction) immune responses for infection diagnosis. It is also important to note that, even outside the Amazonian context, this study appears to be unique for all of Brazil.

Before the actual discussion of the results found in this study, however, it seems interesting to first comment on the eco-epidemiological features of the urban and rural scenarios surveyed that could provide a better understanding of our results. The main feature in the urban scenario is the clustering of families in four separate neighborhoods (Centro, Emerêncio, Vila Nova, and Vila Cruzeiro) within the city of Conceição do Araguaia municipality; the principal feature in the rural scenario, on the other hand, is the territorial dispersal of families into four small villages (Ponta de Terra, São Pedro, Mariahi, and Tracuateua-Trindade) within the Bujaru municipality. In addition, it is important to highlight some distinct ecological factors as possibly facilitating infection transmission–such as their vegetation covers, which can shelter and promote the reproduction of the main vector, *Lutzomyia longipalpis* [10]. It is also worth noting that although at least two-thirds of the urban setting have asphalt pavement, a public water supply, and sewage service, one-third of it is peripheral, with fewer provisions for urban sanitation; the urban area is also cut by a small stream with secondary forest on both sides (with the Emerêncio neighborhood on one side and Vila Nova on the other). Those two neighborhoods showed higher concentrations of infection (230 total cases) (Emerêncio (131) and Vila Nova (99)) as compared to the more urbanized neighborhoods of Centro (78) and Vila Cruzeiro (75) with less forest cover (153 total cases). Sanitary conditions are more precarious in the four small villages in the rural area; however, with no public water supply or sewage service, there are still many pockets of secondary forest surrounding residential buildings. In this scenario, 263 cases of infection were detected: Ponta de Terra (71), São Pedro (43), Mariahi (92), and Tracuateua-Trindade (57).

The total number of infections in the urban setting (383), therefore, significantly exceeded (*p* < 0.05) that of the rural area (263) during the one-year study period. The higher (*p* < 0.05) prevalence and incidence rates of infection in the urban scenario (20.3% and 13.6/100-py respectively) as compared to the rural one (14.1% and 6.8/100-py) suggest that the more aggregated spatial distribution of human families plays a pivotal role in the contiguous distribution of clusters of human infections seen in the four districts of the city of Conceição do Araguaia municipality. Likewise, the sparse spatial distribution of human habitations in the rural scenario is accompanied by a sparse distribution of infections there (except in two neighboring areas, Ponta de Terra and São Pedro, which formed a single cluster). It is also important to consider that a third known risk factor—*L.* (*L.*) *infantum chagasi* cycle in domestic dogs—may also have contributed to higher concentrations of human infections in the urban setting, as a higher (*p* < 0.05) prevalence rate of canine infection was found in that setting (39.2%) as compared to the rural one (32%). Thus, although the study period (one year) seemed small for this clinical-epidemiological approach and was not exactly the same in the two areas studied (urban area 2016–2017 and rural one 2017–2018), we believe that the differences in infection behavior observed in these two areas were, in fact, inherent to the ecological factors and of the social organization of human populations, where a more aggregated spatial distribution of families associated with the highest rate of canine infection (39.2%) in the urban area seemed to have a relevant role in the transmission of infection. However, although feline visceral leishmaniasis caused by *L.* (*L.*) *infantum chagasi* was not the target of this clinical-epidemiological approach, we cannot fail to remember that feline infection, in addition to canine infection, may also represent an important link in the context of the One Health in relation to this anthropozoonotic disease [33]. However, to date, the only case of feline infection in the State of Pará, in the Brazilian Amazon, was related to cutaneous leishmaniasis by *Leishmania* (*L.*) *amazonensis* without any clinical signs of visceral involvement [34].

The present diagnostic approach to human infection allowed us to assess the overall prevalence and incidence rates of infection and provided broad visibility to its clinical-immunological spectrum with five distinct profiles (two symptomatic [SI = AVL and SOI] and three asymptomatic ones [III, SRI, and AI]) [14,15,16]. The research plan made it possible to monitor the evolution of those profiles in endemic areas and make preclinical diagnoses of AVL [28,29]. Few research groups in Brazil have used a similar diagnostic approach to human infections and explored their clinical-epidemiological features, including the prevalence and/or incidence of asymptomatic and symptomatic stages of infection—giving preference instead to diagnostic approaches that use serological tools alone, or cellular approaches, or even both together, without combining results from the same individuals, to allow a broader diagnosis of the immune responses to infection [11,12,35,36,37,38,39,40]. Earlier research dealt only with the epidemiological data of the disease itself (AVL), in urban or rural areas, without considering the epidemiological situation of those infections in the target area surveyed [41,42,43]. The advantage of using the present diagnostic approach to human *L.* (*L.*) *infantum chagasi*-infection was recently confirmed in an epidemiological scenario on the western coast of Honduras, Amapala municipality, in Central America, where systemic infections often cause atypical non-ulcerated cutaneous leishmaniasis (NUCL). The serological (ELISA-IgM/IgG) and cellular (DTH) parameters, in addition to clinical and parasitological examinations, revealed a broad clinical-immunological spectrum of infection consisting of seven profiles: three asymptomatic (Indeterminate Asymptomatic Infection [IAI], Resistant Asymptomatic Infection [RAI], and Final Asymptomatic Infection [FAI]); and four symptomatic (Indeterminate Symptomatic Infection [ISI], Resistant Symptomatic Infection [RSI], Final Symptomatic Infection [FSI], and Early Symptomatic Infection [ESI] [44].

Although Brazil is considered the largest AVL area in Latin America [2] and has a “Visceral Leishmaniasis Control and Surveillance Program” that includes canine serological analysis followed by the euthanasia of seropositive dogs, chemical control of the vector, early diagnosis and treatment of human cases, and population awareness [27], there is still not a workable plan to survey the epidemiological situation of human symptomatic and asymptomatic infections in different endemic regions of that country. Such knowledge would be of great utility for prioritizing areas requiring more urgent AVL control actions.

It is nonetheless possible to judge that the infection prevalence found in the urban scenario (20.3%) is lower than that (61.7%) evidenced by DTH in a periurban area of São José de Ribamar municipality, in Maranhão State but similar to that (19.4%) indicated by ELISA (rk39) [39]. Another urban scenario in northeastern Brazil, in Natal city (Rio Grande do Norte State), showed a 24.6% rate by ELISA (rk39), which was lower than the 38.6% by Lima et al. [38]. The prevalence of infection was evaluated only by ELISA (rk39) in two populations during different periods in Belo Horizonte city (Minas Gerais State), in southeastern Brazil. The first sampling showed 12.9% (2006), 14.7% (2008), and 17.9% (2010) rates; the second sampling period showed 23.7% (2006), 25.6% (2008), and 17% (2010) rates. The results of the first sampling were, therefore, all lower than seen in the present study (20.3%), while those of the second sampling were similar [40].

A cross-analysis in a broader context of prevalence in the rural setting in this study (14.1%) is quite difficult, as there have been very few studies in Brazil that have generated comparable data. One example is a study undertaken in the interior of Jacobina municipality, in Bahia State (northeastern Brazil), with a prevalence of 34.1% (based only on DTH) [11], being clearly higher than that of this study. Two other studies were likewise carried out in northeastern Brazil (in Raposa municipality, Maranhão State), using a soluble extract of *Leishmania* (*L.*) *amazonensis* for DTH and a soluble extract of *L.* (*L.*) *infantum chagasi* for ELISA. The results revealed a prevalence of 18.6% by DTH and 13.5% by ELISA [35,36]; the ELISA finding was similar to that of the present study, while that of DTH was slightly higher. Previous studies by our own group reported infection prevalence in rural areas in northeastern Pará State to be 12.6% in Barcarena municipality [19] and 17% in Cametá municipality [16]—both similar to the infection prevalence found in the present study (14.1%).

The incidence rates of infection in the present study (13.6/100-py in the urban setting and 6.8/100-py in the rural one) are not much different, or even lower, than the general incidence in Brazil. Only a single study reported the incidence in an urban setting in Brazil, in Belo Horizonte city (Minas Gerais State). That study evaluated the incidence only by serology (ELISA-rk39) at three different times, 2006 (14.4%), 2008 (21.1%), and 2010 (11.6%), with rates in the first (2006) and last surveys (2010) similar to that of the present study (13.6/100-py), while the 2008 rate was higher [40]. In terms of incidence in the rural context, estimated to be 6.8/100-py in the present study, we found no records outside of the Brazilian Amazon, and only those found by our own group in northeastern Pará State, 5% in Barcarena municipality [19] and 6.9% in Cametá municipality [16], both similar to those of the present study. In conclusion, we found that both prevalence and incidence rates of infection in an urban setting (20.3% and 13.6/100-py) were significantly higher (*p* < 0.05) than those in a rural setting (14.1% and 6.8/100-py) in the Brazilian Amazon.

With regard to the age distribution of human infections, there is no doubt that the most significant result was that of the ≥21 age group, whose infection rate of 26.6% in the urban setting was higher (*p* < 0.05) than seen in the rural one (19.2%)—evidencing that adult infections in urban area are more frequent, which may be explained by the more recent history of urban setting infections, afflicting individuals of any age, including adults, who had no previous history of infection by *L.* (*L.*) *infantum chagasi*. In contrast, no significant differences were found among the infection rates of the two younger age groups (1–10 and 11–20 years old) or between the urban (12.7% and 20.2%, respectively) and rural (8.6% and 19.8%) settings—indicating that the degree of exposure to infection among individuals aged from 1 to 20 years is similar in both urban and rural settings.

When the distribution of human infections by gender was examined, it was found that male infection rates (26.2% and 20.3%) were higher (*p* < 0.05) than those of females (19.5% and 13.5%) in both urban and rural settings, possibly reflecting male behavior/labors, especially in extra-home environments, that could result in greater exposure to infection. The present research represents the first experience of our group in addressing this variable in an urban setting, although previous study in a rural area carried out some years before found no gender differences in infection rates [19]. It is possible that these variations of infection rates between genders of urban areas in Pará State, in the Brazilian Amazon, may be reflecting the history of those municipalities with more recent settlements (such as Conceição do Araguaia municipality) having male infection rates higher than that of females, while municipalities with older settlements show no gender-related differences.

On the other hand, regarding the prevalence of the clinical-immunological profiles of human infections, there seems to be no doubt that asymptomatic infections represented by AI, SRI, and III profiles (with prevalence rates of 13.4%, 3.4%, and 3.0% in urban area, and 8.3%, 2.6%, and 2.3% in rural area, respectively), represent a significant measure of the degree of human infection in a given endemic area. We, therefore, continue to emphasize the need for the Brazilian AVL Control Program to adopt diagnostic tools to evaluate those variables. It is also important to note that the AI profile (the main representative of the asymptomatic infectious stage) was the most prevalent in both epidemiological scenarios but more prevalent in the urban setting—suggesting that the dynamics of infection transmission in urban settings is faster. No differences were observed, however, between the prevalence of symptomatic profiles SI (=AVL) and SOI in the two areas under analysis (urban (0.2% and 0.4%, respectively) and rural (0.6% and 0.5%, respectively)), which may reflect the relatively short study period of one year.

In terms of the incidences of the clinical-immunological profiles, it was not surprising that, once again, those profiles representing asymptomatic infection (i.e., AI, SRI, and III) had the highest new-infection rates in both scenarios investigated (urban (9.6, 2.2, and 1.6/100 py) and rural (4.3, 1.1 and 1.1/100 py))—evoking again the importance of asymptomatic infection as a risk marker for the onset of AVL (=SI profile). We also cannot fail to note that the AI profile had the highest rate for new infections in both scenarios studied [urban (9.6/100 py) and rural (4.3/100 py)] and that the infection rate in the urban area was twice that of the rural area, confirming the trajectory of AVL urbanization in the Brazilian Amazon (and perhaps throughout Brazil). Once again, differences between the incidence rates of symptomatic profiles (SI (=AVL) and SOI) in the two areas analyzed (urban (0.0 and 0.2/100 py) and rural (0.2 and 0.1/100 py)) may reflect the relatively short duration of our study (one year).

When the prevalence of those infection profiles was analyzed by gender, a higher rate (*p* < 0.05) of AI profile was detected in males (5.0%) than in females (3.3%), but only in the rural area–indicating that infections seem to be more concentrated in males in the rural area due to their gender-specific activities in the extra-home environment where infection transmission is more propitious. In the urban environment, on the other hand, it seems that infection transmission affects both males (6.8%) and females (6.6) equally. When the prevalence of those infection profiles was examined by age group, however, a significant portion of asymptomatic infections (represented by AI, SRI, and III profiles) was seen in the ≥21-year age range in the urban environment, where infection transmission appeared more intense; only AI and SRI profiles composed a significant proportion of asymptomatic infections in the rural area (Table 3). Therefore, as there was a greater concentration of older infection cases in the ≥21-year-old group, it is probable that infection transmission in the rural area is slower than in the urban setting.

The most expressive result regarding the incidence of those infection profiles in terms of gender and age range was that the highest incidence of infection in the AI profile in the ≥21-year range, both in urban (7.4/100-py) and in rural areas (2.9/100-py), showing, once again, that infection transmission in the urban area is more than twice that of the rural area and that, regardless of the environment (urban or rural), new cases of infection occur more frequently in adult individuals (≥21 years old).

The last point worth mentioning refers to the evolution of III profile cases in the two settings investigated. Of the 52 cases diagnosed in the urban setting, only one evolved to the SOI profile within one year; of the 42 III cases diagnosed in the rural setting, only one evolved to the SI profile (=AVL). The diagnosis of only two cases with the genetic potential to evolve to severe forms of infection out of a total of 92 seems, at first, to indicate low resolution, but it must be regarded that those numbers represent only a small sample within a much larger universe of III profile cases that should be investigated within endemic, urban and rural areas. It should also be emphasized that the evolution of the two III cases suggests that those individuals’ immune-genetic susceptibility may have been partial (incomplete) or total (complete). In other words, in those cases in which immune-genetic susceptibility is partial or incomplete, the individual has a moderate clinical form that is usually labeled as a subclinical oligosymptomatic infection (SOI profile), which will spontaneously revert to a cure [25]; in those cases with more intense degrees of susceptibility (total or complete), however, the individuals evolve with strong immunological suppression to the severe AVL (=SI profile) [45,46]. The clinical-immunological evolution of the two III profile cases reported here, one evolving to the SOI profile (which converted to DTH a year later), with the other evolving to acute AVL (which required specific therapy for clinical recovery), are strong examples in that regard.

## 5. Conclusions

Based on the diagnostic approach to human *L.* (*L.*) *infantum chagasi*-infection reported here, it seems clear that AVL urbanization results from a much faster transmission dynamics of infection in the urban area, almost twice as fast as in the rural one, providing that prevalence and incidence rates of infection are significantly higher in the urban area than in the rural one, requiring greater attention on public policies of surveillance, control, and infection prevention. Furthermore, this diagnostic approach not only opens up the possibility of screening for symptomatic and asymptomatic *L.* (*L.*) *infantum chagasi*-infection in endemic areas (urban or rural) but also of monitoring the evolution of their clinical-immunological profiles of infection in any of the epidemiological scenarios involved—making it possible to diagnose and treat early cases (III profile) with genetic susceptibility (IgM reactors) before they evolve into severe form (SI profile = AVL) of infection.

## Figures and Tables

**Figure 1 microorganisms-10-02188-f001:**
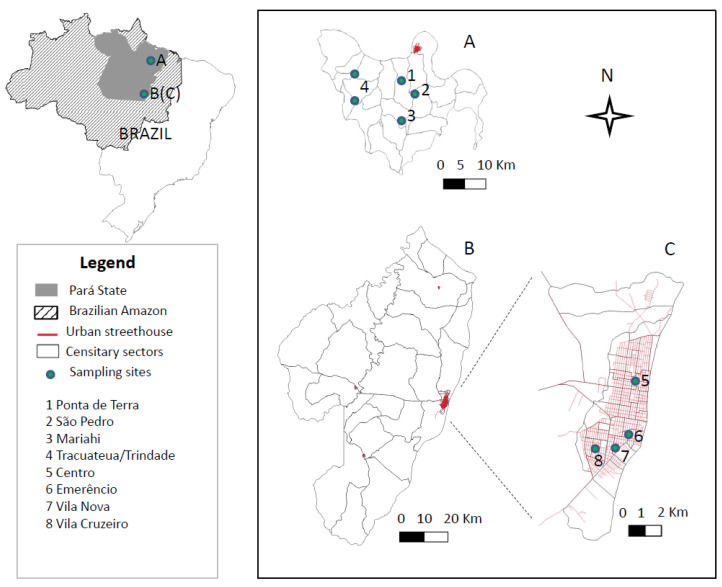
Two distinct epidemiological scenarios in the Brazilian Amazon: the Bujaru municipality in northeastern Pará State (01°30′54″ S: 48°02′41″ W), rural area (**A**), and Conceição do Araguaia municipality, in southeastern Pará State (49°15′53″ S: 49°35′53″ W (**B**), showing the urban area (**C**). The two areas are approximately 900 km apart.

**Figure 2 microorganisms-10-02188-f002:**
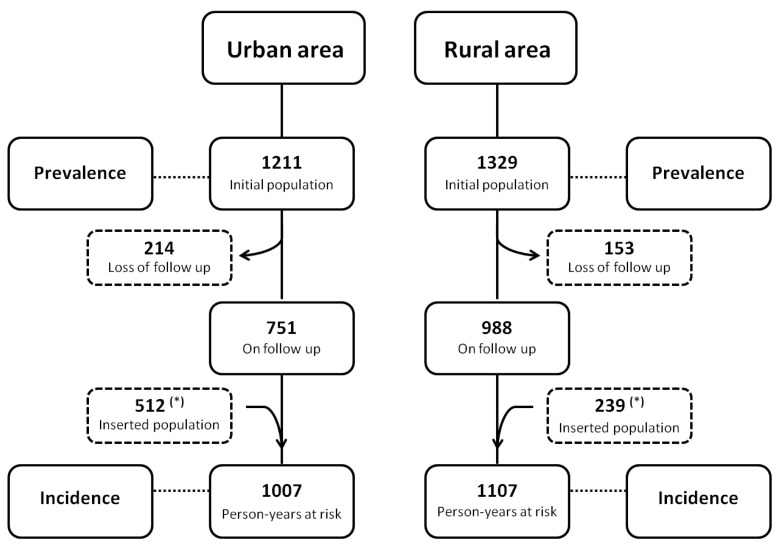
Study population dynamics and study design used to survey prevalence and incidence rates of human *L.* (*L.*) *infantum chagasi*-infection in two distinct epidemiological scenarios in the Brazilian Amazon: the urban area of Conceição do Araguaia municipality and the rural area of Bujaru municipality, Pará State. (*) Individuals who took part in the second stage of the study (incidence survey) as person-years at risk (0.5 person-years at risk).

**Figure 3 microorganisms-10-02188-f003:**
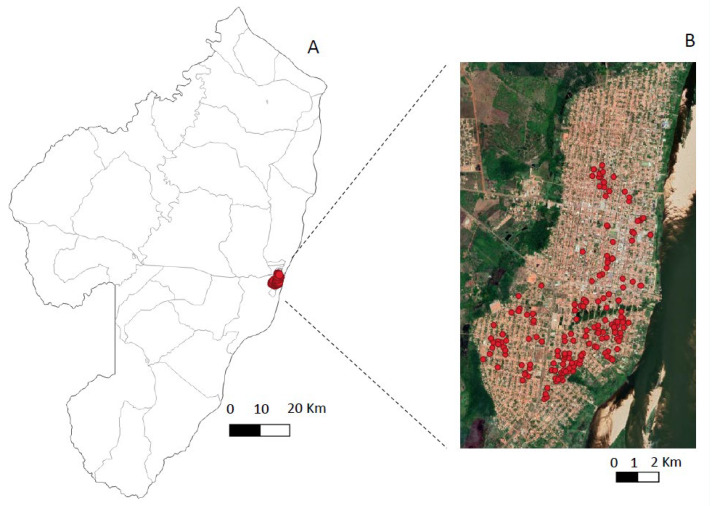
Spatial distribution of positive samples [human *L.* (*L.*) *infantum chagasi*-infections] in Conceição do Araguaia (Pará State, Brazil) (2016–2017). Spatial resolution of the municipality (**A**) and urban area (**B**).

**Figure 4 microorganisms-10-02188-f004:**
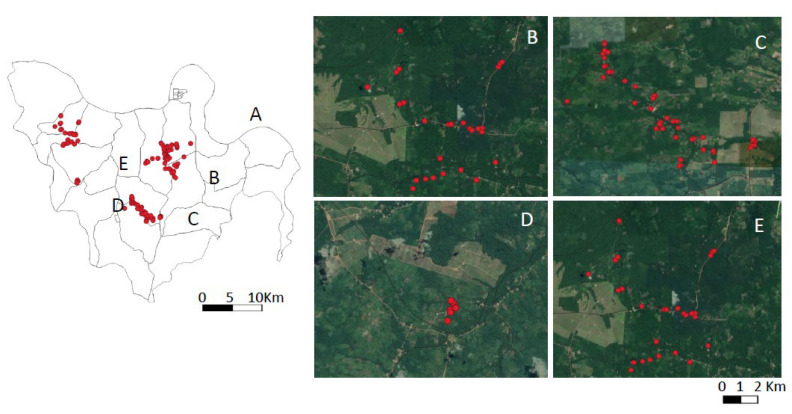
Spatial distribution of positive samples [human *L.* (*L.*) *infantum chagasi*-infections] in Bujaru (Pará State, Brazil) (2017–2018). Spatial resolution of the municipality (**A**) and community areas (**B**–**E**).

**Table 1 microorganisms-10-02188-t001:** Prevalence of the clinical-immunological profiles of human *L.* (*L.*) *infantum chagasi*-infections in distinct urban and rural epidemiological scenarios in Pará State, in the Brazilian Amazon.

Clinical- Immunological Profiles	Urban Scenario *n* = 1211 (%) *n* = 246 (%)	95% CI	Rural Scenario *n* = 1329 (%) *n* = 188 (%)	95% CI
AI *	13.4 (162)	11.5–15.3	8.3 (110)	6.8–9.8
SRI	3.4 (41)	2.4–4.4	2.6 (34)	1.7–3.4
III	3.0 (36)	2.0–3.9	2.3 (30)	1.5–3.1
SOI	0.4 (5)	n/a	0.5 (6)	0.1–0.8
SI	0.2 (2)	n/a	0.6 (8)	0.2–1.0

AI: Asymptomatic Infection; SRI: Subclinical Resistant Infection; III: Indeterminate Initial Infection; SOI: Subclinical Oligosymptomatic Infection; SI (=AVL): Symptomatic Infection. * (*p* < 0.05) n/a: npq < 5, it is not possible to calculate 95% CI.

**Table 2 microorganisms-10-02188-t002:** Incidence of the clinical-immunological profiles of human *L.* (*L.*) *infantum chagasi*-infections in distinct urban and rural epidemiological scenarios in Pará State, in the Brazilian Amazon.

Clinical- Immunological Profiles	Urban Scenario *n* = 1007 (100-py) *n* = 137 (%)	95% CI	Rural Scenario *n* = 1107 (100-py) *n* = 75 (%)	95% CI
AI *	9.6 (97)	7.8–11.5	4.3 (48)	3.1–5.5
SRI	2.2 (22)	1.3–3.1	1.1 (12)	0.5–1.7
III	1.6 (16)	0.8–2.4	1.1 (12)	0.5–1.7
SOI	0.2 (2)	n/a	0.1 (1)	n/a
SI	0.0 (0)	n/a	0.2 (2)	n/a

AI: Asymptomatic Infection; SRI: Subclinical Resistant Infection; III: Indeterminate Initial Infection; SOI: Subclinical Oligosymptomatic Infection; SI (=AVL): Symptomatic Infection. * (*p* < 0.05) n/a: npq < 5, it is not possible to calculate 95% CI.

**Table 3 microorganisms-10-02188-t003:** Prevalence of clinical-immunological profiles of human *L.* (*L.*) *infantum chagasi*-infection by gender and age in distinct urban and rural scenarios in the Brazilian Amazon.

	Gender				Age Range ***					
	Urban (*n* = 1211)		Rural (*n* = 1329)		Urban (*n* = 1211)			Rural (*n* = 1329)		
	Male (%)	Female (%)	Male (%)	Female (%)	1–10 (%)	11–20 (%)	≥21 (%)	1–10 (%)	11–20 (%)	≥21 (%)
AI	6.8 (82)	6.6 (80)	5.0 * (66)	3.3(44)	2.1 (25)	2.6 (31)	8.8 * (106)	0.7 (9)	2.4 * (32)	5.2 * (69)
SRI	1.6 (19)	1.8 (22)	1.6 (21)	1.0 (13)	0.3 (4)	1.2 (14)	1.9 * (23)	0.3 (4)	0.5 (7)	1.7 * (23)
III	1.2 (15)	1.7 (21)	1.1 (15)	1.1 (15)	0.5 (6)	0.5 (6)	2.0 * (24)	0.3 (4)	0.7 (9)	1.3 (17)
SOI	0.2 (2)	0.2 (3)	0.2 (2)	0.3 (4)	0.0 (0)	0.1 (1)	0.3 (4)	0.1 (2)	0.1 (1)	0.2 (3)
SI	0.0 (0)	0.2 (2)	0.4 (5)	0.2 (3)	0.1 (1)	0.0 (0)	0.1 (1)	0.4 (5)	0.1 (1)	0.2 (3)

AI: Asymptomatic Infection; SRI: Subclinical Resistant Infection; III: Indeterminate Initial Infection; SOI: Subclinical Oligosymptomatic Infection; SI (=AVL): Symptomatic Infection. *: *p* < 0.05; ***: years old.

**Table 4 microorganisms-10-02188-t004:** Incidence of clinical-immunological profiles of human *L.* (*L.*) *infantum chagasi*-infection by gender and age in distinct urban and rural scenarios in the Brazilian Amazon.

	Gender				Age Range ***					
	Urban (*n* = 1007)		Rural (*n* = 1107)		Urban (*n* = 1007)			Rural (*n* = 1107)		
	Male (/100-py)	Female (100-py)	Male (100-py)	Female (100-py)	1–10 (100-py)	11–20 (100-py)	≥21 (100-py)	1–10 (100-py)	11–20 (100-py)	≥21 (100-py)
AI	4.2 (42)	5.5 (55)	2.4 (27)	1.9 (21)	0.7 (7)	1.5 (15)	7.4 * (75)	0.2 (2)	1.3 (14)	2.9 * (32)
SRI	1.2 (12)	1.0 (10)	0.7 (8)	0.4 (4)	0.3 (3)	0.6 (6)	1.3 (13)	0.3 (3)	0.1 (1)	0.7 (8)
III	1.0 (10)	0.6 (6)	0.6 (7)	0.5 (5)	0.3 (3)	0.4 (4)	0.9 (9)	0.3 (3)	0.3 (3)	0.5 (6)
SOI	0.0 (0)	0.2 (2)	0.1 (1)	0.0 (0)	0.0 (0)	0.1 (1)	0.1 (1)	0.0 (0)	0.0 (0)	0.1 (1)
SI	0.0 (0)	0.0 (0)	0.2 (2)	0.0 (0)	0.0 (0)	0.0 (0)	0.0 (0)	0.1 (1)	0.1 (1)	0.0 (0)

AI: Asymptomatic Infection; SRI: Subclinical Resistant Infection; III: Indeterminate Initial Infection; SOI: Subclinical Oligosymptomatic Infection; SI (=AVL): Symptomatic Infection. *: *p* < 0.05; ***: years-ol.

## Data Availability

The data that support the findings of this study are available from the corresponding author upon reasonable request.

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
