# Peer review of "Visceral Leishmaniasis Urbanization in the Brazilian Amazon Is Supported by Significantly Higher Infection Transmission Rates Than in Rural Area"

_microorganisms, 2022, doi:10.3390/microorganisms10112188_

Round 1
Reviewer 1 Report
Dear Authors:
The manuscript, "Visceral leishmaniasis urbanization in the Brazilian Amazon is supported by significantly higher infection transmission rates than in rural area", is interesting in that it contributes new data about the epidemiological changes being observed in vector-transmitted diseases.
The manuscript should include more information about the presence of the vector in the studied areas.
The One Health concept should be mentioned and discussed in light of the results obtained, linking it to the data on canine leishmaniasis.
Are there no cases of feline leishmaniasis in the area? In the event that there are reported cases, they should be included in the discussion.
A few formatting issues need to be revised:
The manuscript uses a variety of different fonts.
Parts of the text appear shaded.
The tables have columns that are too narrow for the content, making it difficult to read the information.
The images have a low resolution and look pixelated.
The bibliography needs revising to unify the format.
Author Response
MICROORGANISMS
Reviewer 1
Dear Authors:
The manuscript, "Visceral leishmaniasis urbanization in the Brazilian Amazon is supported by significantly higher infection transmission rates than in rural area", is interesting in that it contributes new data about the epidemiological changes being observed in vector-transmitted diseases.
First of all, let us thank you for your work in reviewing our manuscript submitted to Microorganisms.
The manuscript should include more information about the presence of the vector in the studied areas.
Entomological surveillance was performed on both urban and rural environments, demonstrating intradomiciliary/peridomiciliary presence of the major phlebotomine vector, Lutzomyia longipalpis, in the both environments (unpublished data); however, different sampling efforts applied preclude comparisons between them.
The One Health concept should be mentioned and discussed in light of the results obtained, linking it to the data on canine leishmaniasis.
Ok, it's already been answered in the manuscript discussion.
Are there no cases of feline leishmaniasis in the area? In the event that there are reported cases, they should be included in the discussion.
To date, there are no reports of clinical and/or laboratory diagnosis of feline visceral leishmaniasis by L. (L.) infantum chagasi in any of the studied areas and also in other endemic areas in the State of Pará (northern Brazil). The only report on feline leishmaniasis in the State of Pará concerns a case of cutaneous leishmaniasis caused by Leishmania (L.) amazonensis with no clinical evidence of visceral involvement (Carneiro et al. First report on feline leishmaniasis caused by Leishmania (L.) amazonensis in Amazonian Brazil. Vet Parasitol Reg Stud Reports. 2020 Jan;19:100360. doi: 10.1016/j.vprsr.2019.100360).
A few formatting issues need to be revised:
The manuscript uses a variety of different fonts.
Sorry, we already did a review.
Parts of the text appear shaded.
Sorry, this stemmed from attempts to correct the language. We have tried to remove these inconveniences.
The tables have columns that are too narrow for the content, making it difficult to read the information.
Ok, the required changes have already been fixed.
The images have a low resolution and look pixelated.
The same above.
The bibliography needs revising to unify the format.
The same above.
We hope we have met your comments and requests.

Reviewer 2 Report
The authors decribe in this survey the higher incidence and prevalence of visceral leishmaniasis serology in an urban area in Brazil compared to a rural one. The definition of different groups based on cellular/humoral immune status is an interesting approach.
Below some comments
1. The period of study is not the same for the two areas and relatively short for leishmaniasis for which annual incidence and prevalence may vary significantly in relation with several factors mainly climatic ones. These elements need to be discussed eventhough serologial incidence and prevalence could be less affected by these factors.
2. What is the origin of leishmanin and are there cases of cutaneous leishmaniasis in the the two areas? Can we rule out cross-reactivity with former cutaneous leishmaniasis in asymptomatic patients with positive DTH?
3. Line 143-145 : please explain why these patients were excluded from the study.
4. The specificity of IFAT needs to be discussed especially for the "indetermined initial infection" group.
5. Line 528-533 : what is the relation between the history of the municipality and the sex ratio?
Author Response
MICROORGANISMS Comments and Suggestions for Authors
Reviewer 2
The authors decribe in this survey the higher incidence and prevalence of visceral leishmaniasis serology in an urban area in Brazil compared to a rural one. The definition of different groups based on cellular/humoral immune status is an interesting approach.
First of all, let us thank you for your work in reviewing our manuscript submitted to Microorganisms.
Below some comments
- The period of study is not the same for the two areas and relatively short for leishmaniasis for which annual incidence and prevalence may vary significantly in relation with several factors mainly climatic ones. These elements need to be discussed even though serologial incidence and prevalence could be less affected by these factors.
Allow us to emphasize that, unlike other Brazilian regions (northeast, midwest, southeast and south) the climate in the Brazilian Amazon is not characterized by four distinct seasons, but only two main periods: summer (hot and humid, less rainy) and winter (hot and humid, more rainy), with little expressive variations in temperature and humidity (mainly in the State of Pará), which is why we mentioned it in topic 2.1. Study area (Material and Methods) that “The climate in both is typically equatorial, with a mean temperature of 28º C and high humidity”. Furthermore, although the study period seems small for this type of epidemiological approach and was not exactly the same in the two areas studied, we believe that the differences in infection behavior observed in the two areas are less influenced by these variables, but mainly by the history of infection in the urban area (municipality of Conceição do Araguaia) and in the rural area (municipality of Bujaru), which was also mentioned in the same topic 2.1. Study area, demonstrating that, while AVL is endemic with records for more than thirty years in rural area, the disease only appeared in urban area a little over ten years ago. For this reason, we decided to insert a short paragraph in the “Discussion” to emphasize our interpretation.
- What is the origin of leishmanin and are there cases of cutaneous leishmaniasis in the the two areas? Can we rule out cross-reactivity with former cutaneous leishmaniasis in asymptomatic patients with positive DTH?
In fact, your questioning is quite interesting and pertinent; let's try to clarify. First, as you can see in the references of our group mentioned in the present work (Crescente et al. 2009; Silveira et al. 2009, 2010a,b), the leishmanin used for the intradermal skin test to evidence the delayed-type hypersensitivity (DTH) response has been produced for more than two decades in the "Ralph Lainson" leishmaniasis lab (Parasitology Department of Evandro Chagas Institute) with promastigote antigen (107/ml) of L. (L.) infantum chagasi cultured in RPMI medium. In fact, this laboratory protocol has already been carried out in that laboratory for more than four decades, not only with L. (L.) amazonensis antigen, but also with L. (V.) braziliensis one for the immunodiagnosis of cutaneous leishmaniasis. Today, we are already using axenic cultured amastigote antigen of L. (V.) lainsoni for the diagnosis of cutaneous leishmaniasis. Second, in the Brazilian Amazon, the distribution of visceral and cutaneous leishmaniases is closely coexistent in some areas, although visceral disease is more frequently associated to a peridomestic epidemiology, while cutaneous disease has a strong association to a forest environment. This situation promotes an early L. (L.) infantum chagasi infection in children, which in the great majority of cases (≥95%) results in a long-lasting asymptomatic infection by this parasite due to an efficient and species-specific cell-mediated immunity. On the other hand, cutaneous leishmaniasis has a multiple etiology and is commonly found in adult people working into the forests. Thus, contrary to what you suppose, it is much more likely that the adult individual has already been sensitized as a child with the natural antigen of L. (L.) infantum chagasi, whose formed immunity does not protect him against cutaneous leishmaniasis (Silveira et al. Failure of natural immunity induced by asymptomatic Leishmania (L.) infantum chagasi infection in protecting against cutaneous disease by Leishmania (V.) braziliensis. 30 World/Leish, Italy, 2005).
- Line 143-145 : please explain why these patients were excluded from the study.
You refer to the following sentence (?): Any individuals diagnosed with AVL (=SI profile) and undergoing treatment were excluded from the present study. If so, what we meant is that any individual who already had a previous diagnosis and was being treated for visceral leishmaniasis would not be included in the sample of our study.
- The specificity of IFAT needs to be discussed especially for the "indetermined initial infection" group.
Sorry, but we believe that IFAT specificity has been clearly evidenced from the moment we showed that individuals of the III profile (DTH-/IFAT-IgG+/++) who also showed IFAT-IgM reactivity evolved to AVL (Lima et al. 2014, 2020). That is, although IFAT-IgG reactivity shows a modest response (+/++) in profile III, in those cases truly susceptible to infection the IFAT-IgM reactivity confirmed the early, acute stage of the infection, initially detected by IFAT-IgG reactivity (+/++). In addition, the second paragraph of the topic 2.3. Study design (Material and Methods) also supports this understanding.
- Line 528-533 : what is the relation between the history of the municipality and the sex ratio?
As mentioned before, in the oldest endemic areas (rural area of ​​the municipality of Bujaru) the transmission of the infection affects individuals from children (up to under 1 year old), regardless of gender, reflecting similar infection rates between adult men and women in these areas; while in urban areas with recent transmission of the infection (Conceiçao do Araguaia municipality), the involvement of individuals occurs in different age groups (children, adolescents and adults), however, among adults, men seem to be more exposed than women (perhaps as a result of their work activities in the extra-domestic environment). We hope we have met your comments and requests.